# Changes in Local and Systemic Adverse Effects following Primary and Booster Immunisation against COVID-19 in an Observational Cohort of Dutch Healthcare Workers Vaccinated with BNT162b2 (Comirnaty^®^)

**DOI:** 10.3390/vaccines12010039

**Published:** 2023-12-29

**Authors:** Christiaan Serbanescu-Kele Apor de Zalán, Maud Bouwman, Frits van Osch, Jan Damoiseaux, Mary-Anne Funnekotter-van der Snoek, Frans Verduyn Lunel, Florence Van Hunsel, Janneke de Vries

**Affiliations:** 1Department of Intensive Care, VieCuri Medical Centre, 5912 BL Venlo, The Netherlands; 2Department of Internal Medicine, VieCuri Medical Centre, 5912 BL Venlo, The Netherlands; 3Department of Medical Microbiology, VieCuri Medical Centre, 5912 BL Venlo, The Netherlands; maudbouwman@outlook.com (M.B.); jan.damoiseaux@mumc.nl (J.D.); jdevries@viecuri.nl (J.d.V.); 4Department of Clinical Epidemiology, VieCuri Medical Centre, 5912 BL Venlo, The Netherlands; fvosch@viecuri.nl; 5Department of Epidemiology, NUTRIM School of Nutrition and Translational Research in Metabolism, Maastricht University, 6229 HX Maastricht, The Netherlands; 6Central Diagnostic Laboratory, Maastricht University Medical Centre, 6229 HX Maastricht, The Netherlands; 7Department of Clinical Pharmacy, VieCuri Medical Centre, 5912 BL Venlo, The Netherlands; mfunnekotter@viecuri.nl; 8Department of Medical Microbiology, Utrecht University Medical Centre, 3584 CX Utrecht, The Netherlands; f.m.verduynlunel@umcutrecht.nl; 9Department of Pharmacy, University of Groningen, 9713 AV Groningen, The Netherlands; f.vanhunsel@lareb.nl; 10Netherlands Pharmacovigilance Centre (Lareb), 5237 MH Hertogenbosch, The Netherlands

**Keywords:** COVID-19, vaccine, immunisation, adverse events, healthcare workers

## Abstract

In healthcare workers (HCWs) and in the general population, fear of adverse effects is among the main reasons behind COVID-19 vaccine hesitancy. We present data on self-reported adverse effects from a large cohort of HCWs who underwent primary (N = 470) and booster (N = 990) mRNA vaccination against SARS-CoV-2. We described general patterns in, and predictors of self-reported adverse effect profiles. Adverse effects following immunisation (AEFI) were reported more often after the second dose of primary immunisation than after the first dose, but there was no further increase in adverse effects following the booster round. Self-reported severity of systemic adverse effects was less following booster immunisation. Prior infection with SARS-CoV-2 was found to be a significant predictor of AEFI following primary immunisation, but was no longer a predictor after booster vaccination. Compared to other studies reporting specifically on adverse effects of SARS-CoV-2 vaccination in healthcare workers, we have a relatively large cohort size, and are the first to compare adverse effects between different rounds of vaccination. Compared to studies in the general population, we have a considerably homogenous population. Insights in AEFI following primary and booster vaccinations may help in addressing vaccine hesitancy, both in HCWs and in the general population.

## 1. Introduction

Since its global eruption in 2020, the COVID-19 pandemic has had a marked impact on the lives of people worldwide. Healthcare workers (HCWs) working in direct contact with COVID-19 patients were potentially at high risk of infection. Furthermore, studies have suggested that HCW behaviours during the pandemic may have been different from those in the general population [1]. Apart from personal protective equipment, and basic hygiene measures [2,3], safe and effective vaccines are the primary line of defence against serious COVID-19-related illness, both in the general population, and among healthcare workers [4,5]. One of several approved vaccines against COVID-19, BNT162b2 (Comirnaty^®^; BioNTech, Mainz, Germany) is a nucleoside-modified mRNA vaccine encapsulated in lipid nanoparticles, based on mRNA encoding the SARS-CoV-2 spike protein [6]. It has been shown to induce a strong combined adaptive humoral and cellular immune response [7], and global immunisation, together with several approved vaccines, has led to a considerable decrease in the number of patients presenting with severe COVID-19 [3,4,8]. Waning immunity over time, however, and the emergence of new variants of SARS-CoV-2 has prompted the implementation of additional “booster” doses [9,10,11].

In January 2021, the government of The Netherlands embarked on a nation-wide government immunisation campaign, where priority was given to HCWs who worked in direct patient contact (N = 30,000, nation-wide) in sectors of acute and critical care medicine, both in the ambulance services and in hospitals [12]. In November 2021, all HCWs could obtain a booster vaccination. In accordance with national policy, our centre has participated in both the initial vaccination rounds, offering BNT162b2 to healthcare workers.

Despite active public information campaigns, vaccine hesitancy remains considerable, including among healthcare workers [13,14]. Among healthcare workers, a major reported reason for vaccine hesitancy is fear of adverse effects [11,12]. This creates a necessity to quantify adverse effect profiles, both following initial vaccination, and after subsequent “booster” doses. In a recent Cochrane review analysing different approved vaccine types, serious adverse effects (including anaphylaxis, vaccine-induced immune thrombocytopenia, and myocarditis) were extremely rare, although non-life threatening local and systemic adverse effects were relatively common [15,16]. Adverse effect profiles may, however, be different in specific populations.

In this observational study, we analyse adverse effect profiles in HCWs after vaccination with BNT162b2 in a Dutch teaching hospital. We describe the incidence of different local and systemic AEFI in this population, and compare AEFI profiles after primary and booster immunisation. Finally, we analyse risk factors for local and systemic AEFI.

## 2. Methods

### Study Design and Population

We used observational, retrospective data obtained through questionnaires in our in-hospital vaccination programme. VieCuri Medical Centre is the largest regional hospital in North Limburg, The Netherlands. It has a near full-range of medical specialties, and houses a total of 569 inpatient beds. Among its 3028 employees are around 200 specialist consultants and more than one hundred registrars. VieCuri Medical Centre has a training function, both at the undergraduate and the graduate level, and is a member of the Dutch association of “top-clinical” teaching hospitals (STZ).

At VieCuri Medical Centre, BNT162b2 was offered to HCWs in acute and critical care, from January to February 2021 (Round 1). Selection was according to strict national government guidelines [17], and included only those HCWs with direct patient contact in the ambulance services, in the Emergency Department, on the COVID-19 cohort inpatient wards, and in the Intensive Care Unit. Following informed consent, 2 doses of 30 μg BNT162b2 were administered intramuscularly, with an interval of 21–28 days. From November 2021 to January 2022, all HCWs of VieCuri Medical Centre (not only in acute and critical care) were offered a single booster vaccine (Round 2). After both Round 1 and 2, all HCWs were invited to fill in a questionnaire, with questions on prior SARS-CoV-2 infection, prior vaccine status, and self-perceived severity (mild, moderate, severe) of different local and systemic adverse effects. Participation was fully voluntarily, and questionnaires were filled in anonymously.

Key exclusion criteria for vaccination in Round 1 included recent SARS-CoV-2 infection (<4 weeks), COVID-19-related symptoms at the moment of administration, and planned operation in the next 48 h and pregnancy. After Round 2, exclusion criteria differed slightly. Recent SARS-CoV-2 infection was defined as infection <3 months from immunisation, and pregnancy was no longer an exclusion criterion.

The current study analyses data from individuals who underwent either both doses during Round 1 (N = 720), or booster vaccination during Round 2 (N = 2448). Individuals may have been included in both rounds.

During Round 1, anonymous questionnaires (see Appendix A) were handed out after administration of the second dose. After Round 2, identical questionnaires were handed out (in February 2022). As the survey was taken during the height of the COVID-19 pandemic, at a time when the voluntary nature of vaccination was a subject of hot debate in society, we were not allowed to include any potentially identifying data. Due to the questionnaires being anonymous, we were unable to perform longitudinal comparisons on an individual level (such as paired data analysis). Questions included whether patients had prior confirmed SARS-CoV-2 infection (either by RT-PCR, or by positive serology from a hospital-wide survey in July 2020), including date and self-reported severity of illness, and adverse effects following immunisation (AEFI) after both the first and second dose. We included those AEFI previously observed during the phase 1/2 trial of BNT162b2 in healthy adults^5^, and included local reactions (pain at injection site, injection site swelling or erythema), and systemic symptoms (fatigue, headache, chills, joint pain, fever, and anaphylaxis), and, if present, the self-perceived severity of the reported reaction (mild, moderate, or severe). Questionnaires were collected in an enclosed envelope via internal mail, and data were managed in Castor EDC [18].

The primary outcome studied was the incidence and severity of self-reported local or systemic AEFI, up to a week after administration of either the second dose during Round 1, and after the booster dose during Round 2. We compared incidence and severity of AEFI after Rounds 1 and 2, and after both doses of Round 1. We compared outcomes according to prior confirmed SARS-CoV-2 infection, whether symptomatic or asymptomatic.

All data were extracted and collected from Castor EDC, using IBM SPSS Statistics 25.0 [19] for further analysis. Descriptive statistics were given as counts and percentages. Categorical variables were compared with a χ^2^ test. Odds ratios were calculated with bivariate logistic regression. Our threshold for statistical significance was *p* < 0.05.

In accordance with national and European regulations, we requested and obtained an ethical board approval waiver from the Ethical Board of Maastricht University (METC 2021-3038).

## 3. Results

### 3.1. Baseline Characteristics

During Round 1, 702 HCWs (80.4% of those invited) of VieCuri Medical Centre received two doses of BNT162b2, of whom, in total, 470 HCWs, aged 18–65, filled in the questionnaire after Round 1 (67.0%), of whom one did not complete the survey after the second dose. Prior proven SARS-CoV-2 infection was reported by 135 HCWs (28.8%); infection was symptomatic in 106 of 135 priorly infected HCWs (78.5%), and asymptomatic in 29 (21.5%). During Round 2, a total of 2448 HCWs, out of 3028 employees at VieCuri Medical Centre (80.1%), received a booster vaccine dose. In total, 990 HCWs (40.4% of those vaccinated), aged 18–70 (median 46.5), participated in the current study, of whom 804 were female (81.2%). Prior proven SARS-CoV-2 infection was reported by 449 HCWs (45.4%); infection was symptomatic in 390 of 449 priorly infected HCWs (87.9%), and asymptomatic in 59 (13.3%). Demographic data of both cohorts are shown in Table 1.

### 3.2. Adverse Effects after Round 1

Frequencies of reported adverse effects are shown in Table 2 and Figure 1. After dose 1 of Round 1, 377 participants (80.4%) reported at least one local AEFI, whereas 384 (81.9%) did so after the second dose (χ^2^ = 0.323; *p* = 0.617); similarly, there was no observed difference in the incidence of local AEFI self-perceived by HCWs as severe (4.0% vs. 5.8%; χ^2^ = 1.481; *p* = 0.231). Systemic AEFI were reported more often after the second dose of Round 1, than after the first (61.8% vs. 39.1%; χ^2^ = 48.3, *p* < 0.001), with more HCWs reporting at least one self-perceived severe systemic adverse effect (15.1% vs. 4.7%; χ^2^ = 28.7; p < 0.001). The most commonly reported systemic effects were headache and fatigue.

### 3.3. Adverse Effects after Round 2

In total, 793 out of 990 HCWs (80.1%) reported at least one local AEFI, whereas 566 HCWs (57.2%) reported at least one systemic AEFI. The most commonly reported systemic AEFI were fatigue and headache. Perceived severe systemic AEFI were reported by 103 HCWs (10.4%).

### 3.4. Comparison between Rounds 1 and 2

Frequencies of AEFI, following both doses of Round 1 and following Round 2 are shown in Figure 1. We found no statistically significant difference between the incidence of either any local AEFI after Round 1 and Round 2 (384 or 81.9% vs. 793 or 80.1%; *p* = 0.436; Table 2), or of local AEFI perceived by HCWs as severe (27 or 5.8% vs. 51 or 5.2%; *p* = 0.357). Systemic AEFI were reported by 290 HCWs (61.8%) after Round 1, compared to 566 (57.2%) after Round 2 (*p* = 0.099). Headache, severe fever, and arthralgia were reported more often after Round 1 than after Round 2. Differences in chills, and severe fatigue and headache approached, but did not reach statistical significance. Systemic AEFI perceived by HCWs as severe were reported more frequently after Round 1 than after Round 2 (71 or 15.1% vs. 103 or 10.4%; *p* = 0.012).

### 3.5. Predictors of Adverse Effects after Round 1

After the first dose, there was no difference in the frequency of local AEFI between HCWs with either no prior SARS-CoV-2 infection, prior asymptomatic infection, or prior symptomatic infection (80.2% vs. 72.4% vs. 83.0%; χ^2^ = 1.639; *p* = 0.441). Systemic AEFI were seen more often in HCWs with prior symptomatic SARS-CoV-2 infection (50.0%) than in HCWs with either asymptomatic, or no prior SARS-CoV-2 infection (34.5% and 35.9%, respectively; *p* = 0.031). Frequencies of different systemic AEFI, stratified by prior SARS-CoV-2 infection status, are shown in Figure 2. After dose 1, HCWs with prior SARS-CoV-2 infection reported a higher frequency of fever (11.9% vs. 2.1%, *p* < 0.001), arthralgia (13.3% vs. 5.4%; *p* = 0.004), and chills (13.3% vs. 3.6%; *p* = 0.004), but differences were not statistically significant for headache (27.4% vs. 23.7%; *p* = 0.055) and fatigue (27.4% vs. 20.1%; *p* = 0.055). After dose 2, HCWs with prior symptomatic SARS-CoV-2 infection reported fever and fatigue more frequently. Compared to those who did not, HCWs who reported local AEFI after the first dose were nine times more likely to report these after the second dose (OR 9.42; 95% CI 5.53–16.05); a similar effect was seen for systemic AEFI (OR: 7.14, 95% CI 4.44–11.49). In total, 10 HCWs (2.1%) called in sick after the first dose, compared to 20 HCWs (4.2%) after the second dose (*p* < 0.001). After the first dose, HCWs with previous SARS-CoV-2 infection (N = 135; 28.8%) called in sick more often than HCWs without prior SARS-CoV-2 infection; this difference was not observed after the second dose (8.9% vs. 6.0%, respectively; OR: 1.53, 95% CI 0.73–3.23).

### 3.6. Predictors of AEFI after Round 2

Frequencies of AEFI, stratified by prior SARS-CoV-2 infection status, are shown in Figure 2. After Round 2, prior SARS-CoV-2 infection, symptomatic or asymptomatic, did not correlate with the occurrence of AEFI. Odds ratios for local and systemic AEFI following Round 2 are shown in Figure 3. Prior non-mRNA vaccination was a negative predictor of local AEFI (OR 0.58, 95% CI 0.40–0.86), but not of severe local AEFI, or systemic AEFI. Reported prior side-effects following initial vaccination were correlated with local AEFI (OR 1.93; 95% CI 1.40–2.66), systemic AEFI (1.80; 95% CI 1.378–2.354), and severe systemic AEFI (1.96; 1.20-3.20), whereas the confidence interval for the odds ratio for severe local AEFI crossed 1.0 (OR 1.87; 95% CI 0.95–3.70). Men were less likely to report local AEFI (OR 0.65; 95% CI 0.45–0.95) and systemic AEFI (OR 0.65; 95% CI 0.46–0.89). Time since previous vaccination was not found to be a predictor of any type of AEFI.

## 4. Discussion

We presented retrospective data on adverse events following primary (Round 1, 470 individuals) and booster (Round 2, 990 individuals) mRNA vaccination against COVID-19 in a single-center population of HCWs. Most HCWs experienced at least one local AEFI, whereas between 39.1% and 61.8% of HCWs experienced at least one systemic side effect. Comparing both doses of Round 1, systemic AEFI were reported more often after the second dose, whereas we found no differences in reported local AEFI. Comparing Rounds 1 and 2, we found similar frequencies of local and systemic AEFI, but, importantly, we also found a noticeable decrease in the number of systemic AEFI perceived by HCWs as severe (15.1% vs. 10.4%).

Furthermore, we analysed predictors of AEFI following immunisation against COVID-19. After dose 1 of Round 1, patients with prior SARS-CoV-2 infection, either symptomatic or asymptomatic, were more likely to report systemic AEFI. After dose 2 of Round 1, only patients with prior symptomatic SARS-CoV-2 were more likely to report systemic AEFI. HCWs who reported side effects after dose 1 were up to nine times more likely to report side effects after dose 2. After Round 2, prior proven symptomatic or asymptomatic SARS-CoV-2 infection were no longer found to be a predictor of AEFI. Prior non-mRNA vaccination was correlated with decreased frequency of local AEFI. The male sex was found to be a negative predictor for both local and systemic AEFI, whereas previously reported side effects were a strong positive predictor.

To our knowledge, it is a novel finding that, although absolute frequencies of AEFI were the same after primary and booster vaccinations, the perceived severity of systemic side effects appeared to decrease. This is reassuring, as the occurrence of severe side effects has been reported to influence future vaccine behaviour, and may correlate with subsequent booster vaccine reluctancy [20]. Future, prospective, longitudinal studies would be helpful in further investigating this phenomenon.

Our finding that prior SARS-CoV-2 infection was a positive predictor of AEFI following Round 1 is in line with recent literature [21], which is assumed to be due to a more pronounced immune response to an already-known antigen [22]. A similar mechanism to that of prior infection may underpin the increased frequency of systemic side effects we observed after the second dose of Round 1, compared to the first. To our knowledge, we are the first authors to report that the predictive effect of prior SARS-CoV-2 infection appears to die out with subsequent vaccination. We believe this may be because a previously vaccinated population is no longer ‘antigen-naive’, potentially mitigating differences in immune response between previously infected and non-infected individuals.

Differences in immunological response due to sex, and resultant differences in AEFI patterns have extensively been described for other vaccines, and may be due to differences in both the innate and adaptive immune response [23,24]. The lower incidence of systemic AEFI perceived as severe in men in our population may reflect the same phenomenon. However, data on the impact of biological sex on AEFI following immunisation against COVID-19 is scarce and not completely consistent [21,25,26]. Alternatively, the effect we observed could be due to confounding factors, such as certain sexes being overrepresented in different professional, ethnic, or age subgroups of our study population. Unfortunately, only limited demographic data were available to us, precluding more meaningful analysis of potential confounders.

Several large studies have reported on AEFI related to COVID-19 vaccination in the general population [14,23,27,28]. Compared to these studies, ours has a relatively homogeneous study population, all of whom have been vaccinated with mRNA vaccines. Furthermore, cohorts for both rounds largely overlap, permitting a better comparison of AEFI following primary and booster vaccination.

One of the most important strengths of our study from a public-health perspective is our focus on HCWs. Compared to other studies analysing this population [29,30,31,32], our study has a relatively large sample size. Furthermore, to our knowledge, no other study has compared side effect profiles in HCWs after both doses of primary immunisation, and after booster vaccination. Our study contributes to the ongoing debate on booster vaccination policy. In addition, we believe that a better understanding of AEFI in this specific population may be helpful in addressing HCWs post-vaccine hesitancy. Finally, in a specifically Netherlands-oriented context, our study will help policy makers reflect on adverse effect patterns during this past immunisation campaign.

Our study was limited by several factors. First, our reliance on self-reporting, by definition, allows for a certain measure of subjectivity. On the other hand, self-perceived severity of AEFI has been thought to influence future vaccine behaviour [17], making exactly this subjective piece of data important to analyse. Our second limitation is that Round 1 took place during a nation-wide crisis, with a very short time-span available for planning. Due to overarching concerns about privacy, during a time when free choice to be or not to be vaccinated was a subject of lively public debate, we were not allowed to register any potentially identifying data. This included not only contact information, but also secondary potentially identifying data, such as job title, or, during Round 1, even age and gender. This leads to a relative paucity in demographic data available to us, and, furthermore, means that we have no identifying data allowing us to match questionnaires from Rounds 1 and 2 belonging to the same individuals. This unfortunately precludes us from making subject-specific comparisons and performing paired analyses between Rounds 1 and 2. Furthermore, due to the same reason, we cannot ascertain the amount of overlap between both cohorts. Moreover, we could not ask our HCWs additional questions in retrospect, limiting the amount of demographic data available. Finally, the retrospective and voluntary nature of our questionnaire may have led to selection bias, as subjects’ willingness to fill in the questionnaire may have been influenced by the presence or perceived severity of AEFI.

In conclusion, we described patterns of AEFI among HCWs in a Dutch teaching hospital, and analysed differences in AEFI following primary immunisation and booster vaccination. We found that adverse effects were more frequent after the second dose of primary immunisation. A novel finding was that, after booster vaccination, the frequency of adverse effects remained the same, but the severity of systemic AEFI decreased. Furthermore, although prior SARS-CoV-2 was a positive predictor of AEFI following primary immunisation, it was no longer a predicting factor after booster vaccination. Insights in AEFI following primary and booster vaccinations may help in addressing vaccine hesitancy, both in HCWs and in the general population.

## Figures and Tables

**Figure 1 vaccines-12-00039-f001:**
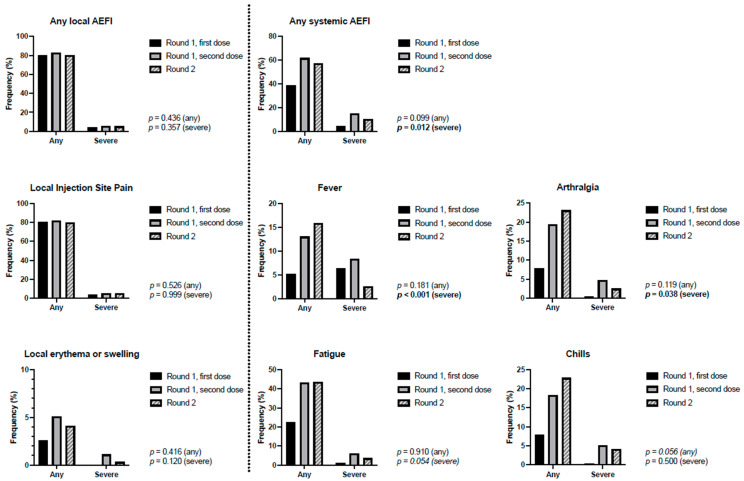
Frequency of adverse effects following immunisation (AEFI) and of AEFI perceived as severe, reported after both doses of Round 1, and following Round 2. Statistical comparison is between second dose of Round 1, and Round 2, and was performed separately per category for any, and severe AEFI. Bold face indicates *p* values under 0.05.

**Figure 2 vaccines-12-00039-f002:**
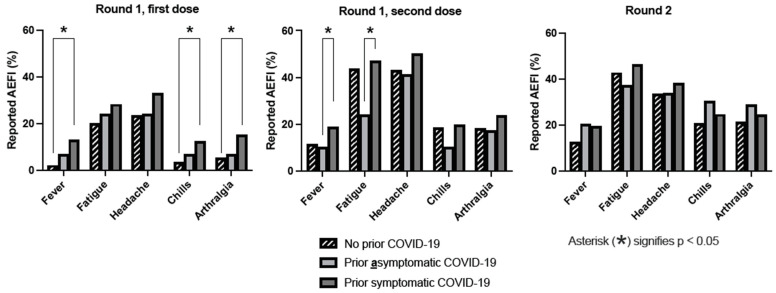
Frequency of AEFI, following both doses of Round 1, and after Round 2, by prior SARS-CoV-2 infection status. Prior symptomatic and asymptomatic infection were reported separately.

**Figure 3 vaccines-12-00039-f003:**
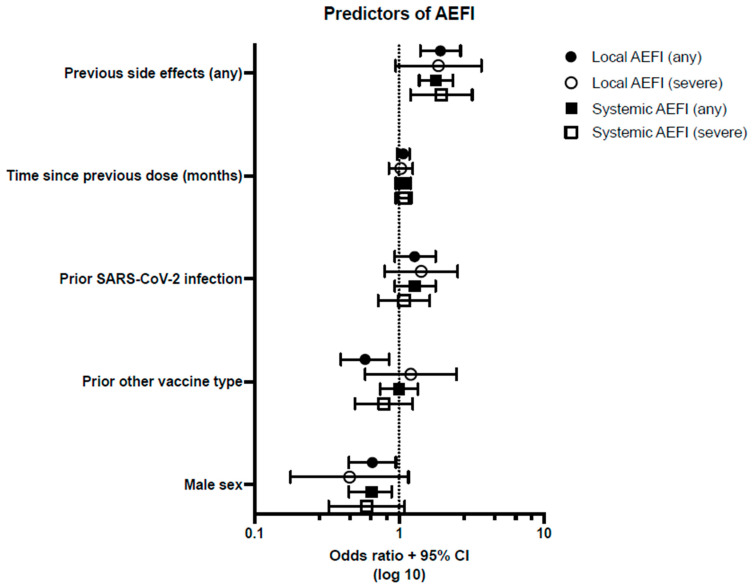
Predictors of adverse effects following immunisation (AEFI), after Round 2, expressed as odds ratios with confidence intervals. The dotted line represents an odds ratio of 1.

**Table 1 vaccines-12-00039-t001:** Demographic data.

	Round 1	Round 2
N vaccinated	702 (80.4% of invited)	2448 (80.1% of all employees)
N of participants in the study	470 (67.0%)	990 (40.4%)
Age	18–65 *	18–70 (median 46.5)
Female sex	Unknown *	804 (81.2%)
Job description	Exclusively HCWs in acute and critical care	All HCWs
Previous proven SARS-CoV-2 infection	135 (28.8%)	449 (45.4%)
Previous vaccine type		
mRNA vaccination (BNT162b2/Comirnaty^®^ or Spikevax-Moderna	None (primary immunisation)	727 (73.4%)
non-mRNA/unknown		263 (26.6%)

* Unfortunately, due to heightened privacy concerns during this phase in the pandemic, at a time when the voluntary nature of vaccination was a subject of hot debate in society, we were not allowed to include any potentially identifying information. During Round 1, policy makers did not allow us to record age and gender; during Round 2, registration of age and gender was allowed, but we were not allowed to note a job description. Age ranges for Round 1 are the extremes of age included.

**Table 2 vaccines-12-00039-t002:** Frequencies of adverse effects following immunisation (AEFI) following both doses of Round 1, and following Round 2. If present, perceived severity of AEFI was reported as mild, moderate, or severe. *p* values indicate statistical comparison between Round 2 and the second dose of Round 1, and are given separately for the presence or absence of AEFI (reported under any), and for the presence or absence of AEFI reported as severe (reported under severe).

	Round 1, First Dose	Round 1, Second Dose	Round 2	*p* (Any) ***	*p* (Severe) ****
** Local AEFI **					
**Injection site pain**				0.526	0.999
*None*	92 (19.6%)	85 (18.1%)	199 (20.1%)		
*Mild*	165 (35.1%)	208 (44.3%)	465 (47.0%)		
*Moderate*	194 (41.3%)	152 (32.4%)	276 (27.9%)		
*Severe*	19 (4.0%)	24 (5.1%)	50 (5.1%)		
**Local erythema or swelling**				0.416	0.120
*None*	458 (97.4%)	445 (94.9%)	949 (95.9%)		
*Mild*	6 (1.3%)	15 (3.2%)	27 (2.7%)		
*Moderate*	6 (1.3%)	4 (0.9%)	11 (1.1%)		
*Severe*	0	5 (1.1%)	3 (0.3%)		
** Systemic AEFI **					
**Fever**				0.181	**<0.001**
*None*	446 (94.9%)	408 (87.0%)	834 (84.2%)		
*Mild*	12 (2.6%)	22 (4.7%)	75 (7.6%)		
*Moderate*	9 (1.9%)	0 (0%)	56 (5.7%)		
*Severe*	3 (6.3%)	39 (8.3%)	25 (2.5%)		
**Fatigue**				0.910	*0.054*
*None*	365 (77.7%)	266 (56.7%)	557 (56.3%)		
*Mild*	61 (13.0%)	91 (19.4%)	239 (24.1%)		
*Moderate*	39 (8.3%)	84 (17.9%)	158 (16.0%)		
*Severe*	5 (1.1%)	28 (6.0%)	36 (3.6%)		
**Headache**				**0.001**	*0.094*
*None*	348 (74.0%)	260 (55.4%)	640 (64.6%)		
*Mild*	62 (13.2%)	84 (17.9%)	181 (18.3%)		
*Moderate*	44 (9.4%)	91 (19.4%)	119 (12.0%)		
*Severe*	16 (3.4%)	34 (7.2%)	50 (5.1%)		
**Arthralgia**				0.119	**0.038**
*None*	433 (92.1%)	378 (80.6%)	761 (76.9%)		
*Mild*	22 (4.7%)	27 (5.8%)	103 (10.4)		
*Moderate*	13 (2.8%)	42 (9.0%)	101 (10.2%)		
*Severe*	2 (0.4%)	22 (4.7%)	25 (2.5%)		
**Chills**				*0.056*	0.500
*None*	433 (92.1%)	383 (81.7%)	764 (77.2%)		
*Mild*	22 (4.7%)	31 (6.6%)	100 (10.1%)		
*Moderate*	13 (2.8%)	31 (6.6%)	84 (8.5%)		
*Severe*	2 (0.4%)	24 (5.1%)	42 (4.2%)		
** Any local AEFI, highest score **				0.436	0.357
*None*	92 (19.6%)	85 (18.1%)	197 (19.9%)		
*Mild*	165 (35.1%)	208 (44.3%)	467 (47.2%)		
*Moderate*	194 (41.3%)	149 (31.8%)	275 (27.8%)		
*Severe*	19 (4.0%)	27 (5.8%)	51 (5.2%)		
** Any systemic AEFI, highest score **				*0.099*	**0.012**
*None*	286 (60.9%)	179 (38.2%)	424 (42.8%)		
*Mild*	94 (20.0%)	99 (21.1%)	243 (24.5%)		
*Moderate*	68 (14.5%)	120 (25.6%)	220 (22.2%)		
*Severe*	22 (4.7%)	71 (15.1%)	103 (10.4%)		
** Called in sick because of AEFI **				0.261	
Yes.	10 (2.1%)	20 (4.3%)	57 (5.8%)		

* Comparing AEFI of any severity after second dose of Round 1 to AEFI after Round 2, on a dichotomous scale (adverse effect present or not). ** Comparing severe AEFI after second dose of Round 1 to AEFI after Round 2 (severe adverse effect present or not). *p* values < 0.05 in bold face; *p* values > 0.05, but <0.1 in italic.

## Data Availability

Data available on request, due to privacy considerations.

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
