# Peer review of "Changes in Local and Systemic Adverse Effects following Primary and Booster Immunisation against COVID-19 in an Observational Cohort of Dutch Healthcare Workers Vaccinated with BNT162b2 (Comirnaty®)"

_vaccines, 2023, doi:10.3390/vaccines12010039_

Round 1
Reviewer 1 Report (Previous Reviewer 1)
Comments and Suggestions for Authors
Dear authors,
I agree with the changes you made to the manuscript in accordance with my comments.
Reviewer 2 Report (Previous Reviewer 2)
Comments and Suggestions for Authors
Authors strongly improved manuscript addressing all my previous comments. It can now be accepted for publication
This manuscript is a resubmission of an earlier submission. The following is a list of the peer review reports and author responses from that submission.
Round 1
Reviewer 1 Report
Comments and Suggestions for Authors
I have several questions and comments about this article.
Small comments:
1) In the title, abstract and text of the manuscript, the authors use the terms "adverse effect" “adverse event”, “side effect” meaning the same event. A common terminology should be adhered to.
2) In the abstract you use the abbreviations "AEFI", "HCW". Please include this abbreviation after the first time the full term appears.
3) Multiple references should be given in single parentheses.
4) line 68 "From November 2021 to January 2022, all HCW ... were offered..." - All, but how much? I would like to see a short description of the VieCuri Medical Center.
5) Table 1.
The name of the table should be given.
More serious remarks and questions:
1) Table 1.
In order to make it easier to understand the large amount of information in the table, it is recommended to slightly reformat the table: introduce an additional vertical column “Presence” or "Manifestation", and then indicate the degree of manifestation: Mild, Moderate, Severe.
2) Please check or explain:
- In the Results, line 107, it is indicated that 469 people participated in the study. In column "Round 1, first dose" of the table, the sum of the various manifestations is 470. If 470 people participated in the first round, then this must be indicated in the text.
- Table 1.
The sum of manifestations in column Round 1, second dose, line Fatigue is 429.
The sum of manifestations in column Round 2 line Any local AEFI, highest score is 991.
3) It is recommended that in the Results section, when percentages are given, absolute numbers should be done. This will make it easier to match the text and table data.
Author Response
R1
Small comments:
- In the title, abstract and text of the manuscript, the authors use the terms "adverse effect" “adverse event”, “side effect” meaning the same event. A common terminology should be adhered to.
We thank the Reviewer for noticing this slight inconsistency in our choice of words. We have adapted the Manuscript accordingly.
- In the abstract you use the abbreviations "AEFI", "HCW". Please include this abbreviation after the first time the full term appears.
We thank the Reviewer for noticing that in our Abstract, we used “AEFI” and “HCW” without first introducing these terms. We have edited the Abstract accordingly. The clarifications in the main body of the text (lines 36 and 88) remain.
- Multiple references should be given in single parentheses.
We have edited the manuscript accordingly
- line 68 "From November 2021 to January 2022, all HCW ... were offered..." - All, but how much? I would like to see a short description of the VieCuri Medical Center.
We agree that a good description of the population to whom vaccination was offered is important to be able to judge better how representative our cohorts were. We have added a brief description of the hospital. We believe both our samples to be representative. During Round 1, we were obliged to follow strict government guidelines on whom to vaccinate (now added to the manuscripts). Of those hospital workers invited, 80.4% took the vaccination. During Round 2, all HCW were invited. The number vaccinated during Round 2 was 80.1% of the entire number of hospital staff at the time. We have edited our manuscript to reflect this more clearly.
5) Table 1.
The name of the table should be given.
We thank the reviewer for noticing this and have added a brief description to the Legends (after the Article text).
(NB. this Table is now called Table 2, as we have also inserted a new table with demographic data as requested).
More serious remarks and questions:
1) Table 1. In order to make it easier to understand the large amount of information in the table, it is recommended to slightly reformat the table: introduce an additional vertical column “Presence” or "Manifestation", and then indicate the degree of manifestation: Mild, Moderate, Severe.
We agree with the Reviewer that Table 1, especially as it lacked a description, may have been a bit difficult to read. We have tried the kind suggestion above, but it made the table extremely large and perhaps even more unwieldy. We have now added a thorough description to the legends, and believe the Table has now become readable. We kindly ask the reviewer to indicate if he agrees; we are of course willing to change the table as requested above, if needed.
2) Please check or explain:
- In the Results, line 107, it is indicated that 469 people participated in the study. In column "Round 1, first dose" of the table, the sum of the various manifestations is 470. If 470 people participated in the first round, then this must be indicated in the text.
We thank the Reviewer for noticing this detail, which had escaped our attention. Indeed, 470 patients participated in the study, whereas one of these 470 patients did not complete the survey after the second dose, leading to a total of 469 in the second round.
- Table 1.
The sum of manifestations in column Round 1, second dose, line Fatigue is 429.
We thank the reviewer for noticing this typo; the number under “none” was supposed to be 266. We have corrected the Table accordingly.
The sum of manifestations in column Round 2 line Any local AEFI, highest score is 991.
We thank the reviewer for noticing this typo; the number under “moderate” was supposed to be 275. We have corrected the Table accordingly.
3) It is recommended that in the Results section, when percentages are given, absolute numbers should be done. This will make it easier to match the text and table data.
We initially refrained from adding absolute numbers to reduce the amount of data mentioned in both the text and the Tables, but we can see how it makes cross-reading the text and the tables easier. We have added absolute numbers to the text, where said numbers also figure in Table 2.

Reviewer 2 Report
Comments and Suggestions for Authors
I was invited to revise the paper entitled "Changes in local and systematic adverse effects following repeated immunisation against COVID-19 in a cohort of Dutch healthcare workers". It was a cohort study aimed to evaluate adverse events after repeated comirnaty vaccination among helathcare workers from a Dutch teaching hospital.
Observations:
- The title should report that this study evaluated only comirnaty vaccine;
- Introduction section is poor. Authors should also describe the vaccination campaign performed in Netherland during the study period;
- Authors should add a table reporting HCW baseline characteristics (age, gender, job, ward of work, history of previous sarscov2 infections, comorbidities);
- Sample size estimation was totally lacking;
- Chi-square test is inappropriate in evaluating paired data;
- It is unknown if patients that performed the third dose were priorly vaccinated with a different vaccine than comirnaty;
- Authors should perform also a multivariate analysis. Patients with previous infection frequently experienced enhanced adverse events;
- Discussion section should be improved. No comparisons with similar studies was performed. Point discussed in lines 222-229 should be analyzed in this paper.
Author Response
R2
I was invited to revise the paper entitled "Changes in local and systematic adverse effects following repeated immunisation against COVID-19 in a cohort of Dutch healthcare workers". It was a cohort study aimed to evaluate adverse events after repeated comirnaty vaccination among helathcare workers from a Dutch teaching hospital.
Observations:
- The title should report that this study evaluated only comirnaty vaccine
We have edited the title as requested.
- Introduction section is poor. Authors should also describe the vaccination campaign performed in Netherland during the study period;
We have edited the section accordingly, and referenced government communications on the vaccination campagin (both in the Introduction and in the Methods). If helpful, we are of course willing to also provide a translation of said documents.
- Authors should add a table reporting HCW baseline characteristics (age, gender, job, ward of work, history of previous sarscov2 infections, comorbidities);
We have added a table as requested. We thank the Reviewer for this suggestion, as it allows readers to visualise better the strengths and weaknesses of our cohort.
- Sample size estimation was totally lacking;
We thank the reviewer for allowing us to elaborate on this point. We agree that a good description of our reference population is important. We believe both our samples to be representative. During Round 1, we were obliged to follow strict government guidelines on whom to vaccinate (now added to the manuscripts). Of those hospital workers invited, 80.4% took the vaccination. During Round 2, all HCW were invited. The number vaccinated during Round 2 was 80.1% of the entire number of hospital staff at the time. We have edited our manuscript to reflect this more clearly.
As for formal post-hoc sample size calculation: this was not performed, as recruiting more personnel was impossible due the nature of the measurements (i.e., an observational point measurement at the time-point of vaccination in the hospital). Moreover, with the expected high rate of return of filled-in anonymous questionnaires we estimated that with 700-900 inclusions, reasonable effect sizes (of 0.25 or smaller) could be shown - for example, comparing those with previous COVID-19 infection (expected 30%-70% distribution and observed 28%-72% in round 1).
- Chi-square test is inappropriate in evaluating paired data;
We thank the Reviewer for this suggestion and allowing us to better clarify the nature of our data. Indeed, the Chi square tests is not appropriate for paired data. Unfortunately, due to the nature of the first vaccination round (in a nation-wide crisis, with a very short planning fase), privacy concerns led our hospital to decide on fully anonymous questionnaires. Even though our cohorts have considerable overlap, this means that we cannot retro-actively match data belonging to the same individual in two rounds. Unfortunately, that means paired analysis has been made impossible.
We have edited the Methods and Discussion to better reflect this limitation.
- It is unknown if patients that performed the third dose were priorly vaccinated with a different vaccine than comirnaty;
Vaccines provided by VieCuri Medical Centre in COVID-19 were indeed exclusively Comirnaty. Still, during our study period, the Dutch immunisation campaign included other vaccine types, including non-mRNA vaccines. During Round 2, we included HCW whose prior vaccination was not during Round 1 in our centre. During Round 2, prior vaccination was with mRNA vaccines in 727 cases (73.4%) and with non-mRNA vaccines or unknown in 263 cases (26.6%).
- Authors should perform also a multivariate analysis. Patients with previous infection frequently experienced enhanced adverse events;
We could not perform a meaningful multivariable analysis because of the paucity of demographic data, which was a direct result of the questionnaires. As our study took place during a very hectic phase of the COVID-19 pandemic, overarching privacy concerns have precluded us from registering potentially identifying data. As an example, age was not known for participants in either round, apart from the known fact that they were between 18-65, as these were the age cohorts included in the vaccination campaign, and also the extremes of age as employed in the sectors in which vaccination was performed. Only for Round 2 did we obtain permission to also register gender. And correcting the one analysis for gender while the other could not would make both results not comparable. We therefore opted to present only crude estimates. No other possible confounders were recorded. We did, however, use previous COVID-19 infection for example to stratify analyses and compare groups based on this variable.
- Discussion section should be improved. No comparisons with similar studies was performed. Point discussed in lines 222-229 should be analyzed in this paper.
We thank the reviewer for this remark, and agree that the relative lack of demographic data is a limitation of our study. We have edited the discussion accordingly.
